# Conservation Laws and Symmetry Reductions of the Hunter–Saxton Equation via the Double Reduction Method

**Molahlehi Charles Kakuli** [1,2,*] , **Winter Sinkala** [1] **and Phetogo Masemola** [2]

1 Department of Mathematical Sciences and Computing, Faculty of Natural Sciences, Walter Sisulu University, Private Bag X1, Mthatha 5117, South Africa; wsinkala@wsu.ac.za
2 School of Mathematics, University of the Witwatersrand, Johannesburg 2000, South Africa; phetogo.masemola@wits.ac.za
* Correspondence: ckakuli@wsu.ac.za; Tel.: +27-047-502-2295

**Abstract:** This study investigates via Lie symmetry analysis the Hunter–Saxton equation, an equation relevant to the theoretical analysis of nematic liquid crystals. We employ the multiplier method to obtain conservation laws of the equation that arise from first-order multipliers. Conservation laws of the equation, combined with the admitted Lie point symmetries, enable us to perform symmetry reductions by employing the double reduction method. The method exploits the relationship between symmetries and conservation laws to reduce both the number of variables and the order of the equation. Five nontrivial conservation laws of the Hunter–Saxton equation are derived, four of which are found to have associated Lie point symmetries. Applying the double reduction method to the equation results in a set of first-order ordinary differential equations, the solutions of which represent invariant solutions for the equation. While the double reduction method may be more complex to implement than the classical method, since it involves finding Lie point symmetries and deriving conservation laws, it has some advantages over the classical method of reducing PDEs. Firstly, it is more efficient in that it can reduce the number of variables and order of the equation in a single step. Secondly, by incorporating conservation laws, physically meaningful solutions that satisfy important physical constraints can be obtained.

**Keywords:** double reduction; Hunter–Saxton equation; lie symmetry analysis; conservation law; invariant solution

## 1. Introduction

In this research article, we focus on the Hunter–Saxton equation, a mathematical model described by the partial differential equation (PDE),

$$(u_t + uu_x)_x = \frac{1}{2}u_x^2, \tag{1}$$

which arises as an Euler–Lagrange equation of a variational principle in the study of a nonlinear wave equation for the director field of a nematic liquid crystal [1]. Equation (1) has attracted significant attention from researchers, prompting numerous studies on it and its derivatives. These investigations have often employed Lie symmetry analysis to explore various properties of the equations and, in certain instances, to uncover solutions.

Nadjafikhah and Ahangari [2] determined the Lie point symmetries of the equation and used the symmetries to find conservation laws and conduct symmetry reductions of the equation. An optimal system of one-dimensional subalgebras of the symmetry algebra of the Hunter–Saxton equation was also constructed. San et al. [3] investigated a modified version of the Hunter–Saxton equation, a third-order nonlinear PDE. Their work featured the utilization of Ibragimov's nonlocal conservation method to derive conservation laws for the equation. Liu and Zhao [4] undertook the study of a generalized two-component Hunter–Saxton system of equations. They determined similarity variables and executed

symmetry reductions for this new generalized system, leading to the discovery of some exact solutions of the system. Yao et al. [5] tackled the periodic Hunter–Saxton equation, introducing a variable coefficient into the generalized equation. They succeeded in finding exact solutions for specific selections of the variable coefficient by employing the classical approach to finding invariant solutions. Johnpillai and Khalique [6] also used Lie symmetry analysis to find exact solutions for yet another generalized version of the Hunter–Saxton equation.

In line with the research outlined above, our study is dedicated to examining the symmetry reductions of the Hunter–Saxton equation, utilizing the double reduction method. Our objectives encompass the identification of Lie point symmetries, the determination of conservation laws through the multiplier method, and the application of the double reduction method to achieve symmetry reductions. This research serves as a valuable addition to the existing body of work on the Hunter–Saxton equation, while also contributing insights into the double reduction method in the search for solutions of PDEs. It must be noted that the double reduction routine we adopt in this article is based on the generalized approach proposed by Bokhari et al. [7], which can be used to study PDEs such as those studied in [8–10], of dimension higher than $1 + 1$.

The double reduction method, introduced by Sjöberg [11,12], is a technique for solving PDEs based on the use of Lie symmetries and conservation laws. For a $(1 + 1)$ PDE of order $q$, the double reduction theory allows for the reduction in the PDE to an ODE of order $q - 1$, provided that the PDE possesses a conservation law and an associated symmetry. Generalizations of the double reduction method have been proposed to handle higher-dimensional PDEs and systems of PDEs [7,13,14]. Anco and Gandarias [15] have introduced a further generalization of the double reduction method to handle partial differential equations (PDEs) with $n \geq 2$ independent variables and a symmetry algebra of dimension at least $n - 1$. In their work [15], they present an algorithm for identifying all symmetry-invariant conservation laws that reduce to first integrals for the corresponding ordinary differential equation (ODE) governing symmetry-invariant solutions of the PDE.

Moreover, Anco and Gandarias [15] propose an improved formulation for assessing the symmetry invariance of conservation laws by utilizing multipliers. This refined formulation enables the direct derivation of symmetry-invariant conservation laws, eliminating the need to first obtain conservation laws and subsequently verify their invariance.

The subsequent sections of this paper are structured as follows: Section 2 provides an overview of the necessary preliminaries and outlines the fundamental principles of the double reduction theorem. In Section 3, we calculate the Lie point symmetries and conservation laws for the Hunter–Saxton equation, determining which conservation laws are associated with symmetries. Section 4 focuses on executing symmetry reductions for the Hunter–Saxton equation. Finally, in Section 5, we present our concluding remarks.

## 2. Fundamentals of the Double Reduction Theorem

In this section, we present the double reduction routine for a $q$th-order $(q \geq 1)$ partial differential equation with $n$ independent variables $x = (x^1, x^2, \ldots, x^n)$ and one dependent variable $u = u(x)$, namely

$$F(x, u, u_{(1)}, u_{(2)}, \ldots, u_{(q)}) = 0, \tag{2}$$

where $u_{(q)}$ denotes the collection $\{u_q\}$ of $q$th-order partial derivatives. In this connection, we first present the following well-known definitions and results (see, e.g., [7,16–19]).

1.  The total derivative operator with respect to $x^i$ is

$$D_i = \frac{\partial}{\partial x^i} + u_i \frac{\partial}{\partial u} + u_{ij} \frac{\partial}{\partial u_j} + \cdots, \quad i = 1, 2, \ldots, n, \tag{3}$$

where $u_i$ denotes the derivative of $u$ with respect to $x^i$. Similarly, $u_{ij}$ denotes the derivative of $u$ with respect to $x^i$ and $x^j$.

2. An $n$-tuple $T = \left(T^1, T^2, \ldots, T^n\right), \quad i = 1, 2, \ldots, n$, such that

$$D_i T^i = 0 \tag{4}$$

holds for all solutions of (2) is known as a conservation law of (2).

3. Multiplier $\Lambda$ for Equation (2) is a non-singular function on the solution space of (2) with the property

$$D_i T^i = \Lambda E \tag{5}$$

for arbitrary function $u\left(x^1, x^2, \ldots, x^n\right)$.

4. The determining equations for multipliers are obtained by taking the variational derivative

$$\frac{\delta}{\delta u}(\Lambda E) = 0, \tag{6}$$

where the Euler operator $\delta/\delta u$ is defined by

$$\frac{\delta}{\delta u} = \frac{\partial}{\partial u} - D_i \frac{\partial}{\partial u_i} + D_{ij} \frac{\partial}{\partial u_{ij}} - D_{ijk} \frac{\partial}{\partial u_{ijk}} + \cdots . \tag{7}$$

5. A Lie symmetry of (2) with infinitesimal generator $X = \xi_i \partial x_i + \eta \partial u$ is said to be associated with a conserved law (4) if the symmetry and the conservation law satisfy the relations [16]

$$\left[T^i, X\right] = X\left(T^i\right) + T^i D_j \xi^j - T^j D_j \xi^i, \quad i = 1, \ldots, n. \tag{8}$$

Suppose that the PDE (2) admits a Lie point symmetry with infinitesimal generator $X = \xi_i \partial x_i + \eta \partial u$ that is associated with a conservation law $D_i T^i = 0$. The following steps constitute the routine of the double reduction method:

I. Find similarity variables $\tilde{x}_i, i = 1, 2, \ldots, n$ and $w$,

$$\tilde{x}_i = \tilde{x}_i\left(x^1, x^2, \ldots, x^n\right), \quad i = 1, 2, \ldots, n$$
$$w(\tilde{x}_1, \ldots, \tilde{x}_{n-1}) = \omega\left(x^1, x^2, \ldots, x^n\right) u,$$

such that in these variables $X = \dfrac{\partial}{\partial \tilde{x}_n}$.

II. Find inverse canonical coordinates

$$x^i = x^i(\tilde{x}_1, \tilde{x}_2, \ldots, \tilde{x}_n), \quad i = 1, 2, \ldots, n$$
$$u\left(x^1, x^2, \ldots, x^n\right) = \psi(\tilde{x}_1, \tilde{x}_2, \ldots, \tilde{x}_n) w.$$

III. Write partial derivatives of $u$ in terms of the similarity variables.

IV. Construct matrices $A$ and $A^{-1}$ as follows:

$$A = \begin{pmatrix} \widetilde{D}_1 x_1 & \widetilde{D}_1 x_2 & \ldots & \widetilde{D}_1 x_n \\ \widetilde{D}_2 x_1 & \widetilde{D}_2 x_2 & \ldots & \widetilde{D}_2 x_n \\ \vdots & \vdots & \vdots & \vdots \\ \widetilde{D}_n x_1 & \widetilde{D}_n x_2 & \ldots & \widetilde{D}_n x_n \end{pmatrix}, \quad A^{-1} = \begin{pmatrix} D_1 \tilde{x}_1 & D_1 \tilde{x}_2 & \ldots & D_1 \tilde{x}_n \\ D_2 \tilde{x}_1 & D_2 \tilde{x}_2 & \ldots & D_2 \tilde{x}_n \\ \vdots & \vdots & \vdots & \vdots \\ D_n \tilde{x}_1 & D_n \tilde{x}_2 & \ldots & D_n \tilde{x}_n \end{pmatrix}.$$

V.  Write components $T^i$ of the conserved vector in terms of the similarity variables as follows:

$$\begin{pmatrix} \widetilde{T}^1 \\ \widetilde{T}^2 \\ \vdots \\ \widetilde{T}^n \end{pmatrix} = J\left(A^{-1}\right)^T \begin{pmatrix} T^1 \\ T^2 \\ \vdots \\ T^n \end{pmatrix}, \tag{9}$$

where $J = \det(A)$. Note that $T^1, \ldots, T^n$ in (9) are easily expressed in terms of the similarity variables in light of II and III.

VI.  The reduced conservation law becomes

$$D_1\widetilde{T}^1 + D_2\widetilde{T}^2 + \cdots + D_{n-1}\widetilde{T}^{n-1} = 0. \tag{10}$$

## 3. Symmetries and Conservation Laws of the Hunter–Saxton Equation

The Hunter–Saxton Equation (1) is a $(1+1)$ PDE with two independent variables $x = (x^1, x^2) = (t, x)$ and one dependent variable $u = u(t, x)$. It admits the following four symmetries:

$$X_1 = x\frac{\partial}{\partial x} + u\frac{\partial}{\partial u} \qquad X_2 = \frac{\partial}{\partial t}$$
$$X_3 = t\frac{\partial}{\partial t} + x\frac{\partial}{\partial x} \qquad X_4 = t^2\frac{\partial}{\partial t} + 2tx\frac{\partial}{\partial x} + 2x\frac{\partial}{\partial u}. \tag{11}$$

The symmetries are easily computed using MathLie, the symmetry-finding package for Mathematica [20] developed by G. Baumann [21]. We use the multiplier approach to derive conservation laws for the Hunter–Saxton Equation (1). We seek first-order multipliers

$$\Lambda = \Lambda(x, t, u, u_x, u_t) \tag{12}$$

of (1), for which the determining equation according to (6) is

$$\frac{\delta}{\delta u}\left[\Lambda\left((u_t + uu_x)_x - \frac{1}{2}ux^2\right)\right] = 0, \tag{13}$$

where the standard Euler operator $\delta/\delta u$, as defined in (7), is

$$\frac{\delta}{\delta u} = \frac{\partial}{\partial u} - D_t\frac{\partial}{\partial u_t} - D_x\frac{\partial}{\partial u_x} + D_t^2\frac{\partial}{\partial u_{tt}} + D_x^2\frac{\partial}{\partial u_{xx}} + D_xD_t\frac{\partial}{\partial u_{tx}} - \cdots, \tag{14}$$

and total derivative operators $D_t$ and $D_x$ using (3) are

$$D_t = \frac{\partial}{\partial t} + u_t\frac{\partial}{\partial u} + u_{tt}\frac{\partial}{\partial u_t} + u_{tx}\frac{\partial}{\partial u_x} + \cdots,$$
$$D_x = \frac{\partial}{\partial x} + u_x\frac{\partial}{\partial u} + u_{xx}\frac{\partial}{\partial u_x} + u_{tx}\frac{\partial}{\partial u_t} + \cdots.$$

The determining equation for the multiplier $\Lambda$ after expansion takes the following form:

$$\Omega_0 + u_{tt}\Omega_1 + u_{tx}\Omega_2 + (u_{tx})^2\Omega_3 + u_{xx}\Omega_4 + u_{xx}u_{tt}\Omega_5 = 0, \tag{15}$$

where

$$
\begin{aligned}
\Omega_0 &= u_x\Lambda_{tu} - \frac{1}{2}u_x^2\Lambda_{tu_t} + \Lambda_{tx} - \frac{1}{2}u_x^3\Lambda_{uu_x} - \frac{1}{2}u_x^2u_t\Lambda_{uu_t} + uu_x^2\Lambda_{uu} + u_xu_t\Lambda_{uu} \\
&\quad + 2uu_x\Lambda_{xu} + u_t\Lambda_{xu} + u\Lambda_{xx} - \frac{1}{2}u_x^2\Lambda_{xu_x} + \frac{3u_x^2\Lambda_u}{2} + u_x\Lambda_x, \\
\Omega_1 &= u_x\Lambda_{uu_t} - \frac{1}{2}u_x^2\Lambda_{u_tu_t} + \Lambda_{xu_t}, \\
\Omega_2 &= 2uu_x\Lambda_{uu_t} + 2u\Lambda_{xu_t} - u_x^2\Lambda_{u_tu_x} + 2\Lambda_u, \\
\Omega_3 &= u\Lambda_{u_tu_t} - \Lambda_{u_tu_x}, \\
\Omega_4 &= \Lambda_{tu_x} - u\Lambda_{tu_t} + u_t\Lambda_{uu_x} + u\Lambda_{xu_x} + uu_x\Lambda_{uu_x} - uu_t\Lambda_{uu_t} - \frac{1}{2}u_x^2\Lambda_{u_xu_x} \\
&\quad + 2u\Lambda_u - u_x\Lambda_{u_x} - u_t\Lambda_{u_t} + \Lambda, \\
\Omega_5 &= \Lambda_{u_tu_x} - u\Lambda_{u_tu_t}.
\end{aligned}
$$

The multiplier determining Equation (15) splits with respect to different combinations of the derivatives $u_{xx}$, $u_{tx}$ and $u_{tt}$ yielding an overdetermined linear system of equations for the multiplier. The system of equations was solved using Mathematica [20] to obtain

$$
\Lambda = u_t\left(\delta_2 + \delta_3 t - \frac{\delta_1 t^2}{2}\right) + u_x x(\delta_3 - \delta_1 t) + \delta_1 x + \delta_4 u_x + \frac{\delta_5}{u_x^2}, \tag{16}
$$

where $\delta_i$, $i = 1, 2, \ldots, 5$, are arbitrary constants. From (5) and (16), we obtain

$$
\left[(u_t + uu_x)_x - \frac{1}{2}ux^2\right]\left[u_t\left(\delta_2 + \delta_3 t - \frac{\delta_1 t^2}{2}\right) + u_x x(\delta_3 - \delta_1 t) \right.
$$
$$
\left. + \delta_1 x + \delta_4 u_x + \frac{\delta_5}{u_x^2}\right] = D_t T^t + D_x T^x, \tag{17}
$$

where

$$
\begin{aligned}
T^t &= u_x^2\left(u\left(\frac{\delta_1 t^2}{4} - \frac{\delta_2}{2} - \frac{\delta_3 t}{2}\right) + x\left(\frac{\delta_3}{2} - \frac{\delta_1 t}{2}\right) + \frac{\delta_4}{2}\right) - \frac{\delta_5}{u_x} + \phi_2(x) \\
&\quad + u_x(\delta_1 x - \delta_1 t u + \phi_1(u)), \\
T^x &= u_t^2\left(\frac{\delta_2}{2} + \frac{\delta_3 t}{2} - \frac{\delta_1 t^2}{4}\right) + uu_x^2\left(x\left(\frac{\delta_3}{2} - \frac{\delta_1 t}{2}\right) + \frac{\delta_4}{2}\right) - \frac{\delta_5 u}{u_x} + \frac{3\delta_5 x}{2} \\
&\quad + u_x\left(uu_t\left(\delta_2 + \delta_3 t - \frac{\delta_1 t^2}{2}\right) + \delta_1 ux\right) + u_t(\delta_1 t u - \phi_1(u)) + \phi_3(t)
\end{aligned}
$$

for arbitrary functions $u(t, x)$. When $u(t, x)$ is a solution of Equation (1), the left hand side of (17) vanishes and we obtain conservation laws of the Hunter–Saxton Equation (1) for which the conserved vectors $(T_i^1, T_i^2)$, $i = 1, 2, \ldots, 5$, are given by

$$
\begin{aligned}
T_1^1 &= u_x\left(ux - \frac{1}{2}t^2uu_t\right) - \frac{t^2u_t^2}{4} - \frac{1}{2}tuu_x^2 x + u_t(tu - \phi_1(u)) + \phi_3(t), \\
T_1^2 &= u_x^2\left(\frac{t^2 u}{4} - \frac{tx}{2}\right) + u_x(x - tu + \phi_1(u)) + \phi_2(x),
\end{aligned}
$$

$$
\begin{aligned}
T_2^1 &= \phi_3(t) + uu_xu_t - u_t\phi_1(u) + \frac{u_t^2}{2}, \\
T_2^2 &= u_x\phi_1(u) + \phi_2(x) - \frac{uu_x^2}{2},
\end{aligned}
$$

$$T_3^1 = tuu_xu_t + \frac{tu_t^2}{2} + \phi_3(t) + \frac{1}{2}uu_x^2x - u_t\phi_1(u),$$

$$T_3^2 = u_x^2\left(\frac{x}{2} - \frac{tu}{2}\right) + u_x\phi_1(u) + \phi_2(x),$$

$$T_4^1 = \phi_3(t) - u_t\phi_1(u) + \frac{uu_x^2}{2},$$

$$T_4^2 = \phi_2(x) + u_x\phi_1(u) + \frac{u_x^2}{2},$$

$$T_5^1 = \phi_3(t) - u_t\phi_1(u) - \frac{u}{u_x} + \frac{3x}{2},$$

$$T_5^2 = \phi_2(x) + u_x\phi_1(u) - \frac{1}{u_x}.$$

According to (8), symmetry $X$ is associated with conservation law $D_tT^t + D_xT^x = 0$ if the following formula is satisfied:

$$X\begin{pmatrix} T^t \\ T^x \end{pmatrix} - \begin{pmatrix} D_t\xi^t & D_x\xi^t \\ D_t\xi^x & D_x\xi^x \end{pmatrix}\begin{pmatrix} T^t \\ T^x \end{pmatrix} + (D_t\xi^t + D_x\xi^x)\begin{pmatrix} T^t \\ T^x \end{pmatrix} = 0. \quad (18)$$

It turns out that the association of symmetries and conservation laws of (1) is obtained in the following cases:

$$\kappa_1(X_1 + 2X_3) + \kappa_2X_2 \rightarrow \begin{cases} T_2^1 = \frac{u_t^2}{2} - \frac{\delta_1u_t}{u} + uu_xu_t \\ T_2^2 = \frac{\delta_1u_x}{u} + \frac{\delta_3}{x} - \frac{uu_x^2}{2} \end{cases},$$

$$\kappa_1(X_1 + X_3) + \kappa_2X_2 \rightarrow \begin{cases} T_4^1 = \frac{uu_x^2}{2} - \frac{\delta_1u_t}{u} \\ T_4^2 = \frac{\delta_1u_x}{u} + \frac{\delta_3}{x} + \frac{u_x^2}{2} \end{cases},$$

$$\kappa_1\left(X_1 - \frac{X_3}{2}\right) + \kappa_2X_2 \rightarrow \begin{cases} T_5^1 = \frac{\delta_2}{2\kappa_2 - \kappa_1t} - \frac{\delta_1u_t}{u} - \frac{u}{u_x} + \frac{3x}{2} \\ T_5^2 = \frac{\delta_1u_x}{u} + \frac{\delta_3}{x} - \frac{1}{u_x} \end{cases},$$

$$X_3 \rightarrow \begin{cases} T_3^1 = \frac{\delta_1}{t} + tuu_xu_t + \frac{tu_t^2}{2} + \frac{1}{2}uu_x^2x - u_t\phi_1(u) \\ T_3^2 = \frac{\delta_2}{x} + u_x^2\left(\frac{x}{2} - \frac{tu}{2}\right) + u_x\phi_1(u) \end{cases}.$$

It is important to observe that among the five computed conservation laws, we identified associated Lie point symmetries for only four. Notably, the conservation law $T_1$ lacks any associated Lie point symmetry of the Hunter–Saxton equation.

## 4. Double Reduction of the Hunter–Saxton Equation

*4.1. Double Reduction of (1) by $\langle \kappa_1(X_1 + 2X_3) + \kappa_2X_2 \rangle$*

We transform the generator $Z = \kappa_1(X_1 + 2X_3) + \kappa_2X_2$ to its canonical form $Y = 0\frac{\partial}{\partial r} + \frac{\partial}{\partial s} + 0\frac{\partial}{\partial w}$. Therefore, canonical coordinates $r = r(t, x), s = s(t, x)$ and $w = w(t, x, u)$ must be found such that $Z(r) = 0, Z(s) = 1$ and $Z(w) = 0$. While the coordinates $r$ and $w$ are obtained from invariants of $Z$, the coordinate $s$ may be determined by inspection. More systematically, it can be obtained from an invariant $J = v - s(x, y)$ of the extended operator $Z + \partial_v$, where $v$ is an auxiliary variable [19]. We obtain

$$r = \frac{x}{(2\kappa_1t + \kappa_2)^{3/2}}, \quad s = \frac{\ln x}{3\kappa_1}, \quad w = \frac{u}{\sqrt{2\kappa_1t + \kappa_2}}, \quad \kappa_1 \neq 0, \quad (19)$$

where $w = w(r)$. Inverse canonical coordinates follow from (19) and are given by

$$t = \frac{e^{2\kappa_1s} - \kappa_2r^{2/3}}{2\kappa_1r^{2/3}}, \quad x = e^{3\kappa_1s}, \quad u = \frac{we^{\kappa_1s}}{r^{1/3}}. \quad (20)$$

Computing $A$ and $\left(A^{-1}\right)^T$, we obtain

$$A = \begin{pmatrix} D_r t & D_r x \\ D_s t & D_s x \end{pmatrix} = \begin{pmatrix} -\dfrac{e^{2\kappa_1 s}}{3\kappa_1 r^{5/3}} & 0 \\ \dfrac{e^{2\kappa_1 s}}{r^{2/3}} & 3e^{3\kappa_1 s}\kappa_1 \end{pmatrix}$$

and

$$\left(A^{-1}\right)^T = \begin{pmatrix} D_t r & D_x r \\ D_t s & D_x s \end{pmatrix} = \begin{pmatrix} -3e^{-2\kappa_1 s}\kappa_1 r^{5/3} & e^{-3\kappa_1 s} r \\ 0 & \dfrac{e^{-3\kappa_1 s}}{3\kappa_1} \end{pmatrix}.$$

The partial derivatives of $u$ from (20) are given by

$$
\begin{aligned}
u_t &= \kappa_1 \sqrt[3]{r}\, e^{-\kappa_1 s}(w - 3rw_r), \quad u_x = r^{2/3} w_r e^{-2\kappa_1 s}, \\
u_{tx} &= -\kappa_1 r^{4/3} e^{-4\kappa_1 s}(3rw_{rr} + 2w_r), \\
u_{xx} &= r^{5/3} w_{rr} e^{-5\kappa_1 s}.
\end{aligned}
\tag{21}
$$

The reduced conserved form is given by

$$\begin{pmatrix} T_2^r \\ T_2^s \end{pmatrix} = J\left(A^{-1}\right)^T \begin{pmatrix} T_2^t \\ T_2^x \end{pmatrix}, \tag{22}$$

where $J = \det(A) = -\dfrac{e^{5\kappa_1 s}}{r^{5/3}}$. By substituting (20) and (21) into (22), we obtain

$$
\begin{aligned}
T_2^r &= \delta_1 \kappa_1 + 3\delta_3 \kappa_1 + 3\kappa_1^2 r w w_r - \frac{9}{2}\kappa_1^2 r^2 w_r^2 - \frac{\kappa_1^2 w^2}{2} + \frac{3}{2}\kappa_1 r w w_r^2 - \kappa_1 w^2 w_r, \\
T_2^s &= w_r\left(\kappa_1 w - \frac{\delta_1}{w} - \frac{w^2}{3r}\right) + \frac{\delta_1}{3r} - \frac{\kappa_1 w^2}{6r} + w_r^2\left(w - \frac{3\kappa_1 r}{2}\right),
\end{aligned}
\tag{23}
$$

where the reduced conserved form satisfies

$$D_r T_2^r = 0. \tag{24}$$

From (23) and (24), we have

$$3\kappa_1^2 r w w_r - \frac{9}{2}\kappa_1^2 r^2 w_r^2 - \frac{\kappa_1^2 w^2}{2} + \frac{3}{2}\kappa_1 r w w_r^2 - \kappa_1 w^2 w_r = k,$$

where $k$ is an arbitrary constant.

*4.2. Double Reduction of (1) by $\langle \kappa_1(X_1 + X_3) + \kappa_2 X_2 \rangle$*

Canonical coordinates determined from $\langle \kappa_1(X_1 + X_3) + \kappa_2 X_2 \rangle$ are

$$r = \frac{x}{(\kappa_1 t + \kappa_2)^2}, \quad s = \frac{\ln x}{2\kappa_1}, \quad w = \frac{u}{\sqrt{x}}, \quad \kappa_1 \neq 0, \tag{25}$$

where $w = w(r)$, and the inverse canonical coordinates are given by

$$t = -\frac{\kappa_2 \sqrt{r} - e^{\kappa_1 s}}{\kappa_1 \sqrt{r}}, \quad x = e^{2\kappa_1 s} \quad u = w e^{\kappa_1 s}. \tag{26}$$

Therefore, the partial derivatives of $u$ from (26) are given by

$$
\begin{aligned}
u_t &= -2\kappa_1 r^{3/2} w_r, \quad u_x = \frac{1}{2} e^{-\kappa_1 s}(2rw_r + w), \\
u_{tx} &= -e^{-2\kappa_1 s}\kappa_1 r^{3/2}(2rw_{rr} + 3w_r, \\
u_{xx} &= -\frac{1}{4} e^{-3\kappa_1 s}(w - 4r(rw_{rr} + w_r)).
\end{aligned}
\tag{27}
$$

As for $A$ and $\left(A^{-1}\right)^T$, we obtain

$$A = \begin{pmatrix} D_r t & D_r x \\ D_s t & D_s x \end{pmatrix} = \begin{pmatrix} -\frac{e^{\kappa_1 s}}{2\kappa_1 r^{3/2}} & 0 \\ \frac{e^{\kappa_1 s}}{\sqrt{r}} & 2e^{2\kappa_1 s}\kappa_1 \end{pmatrix},$$

and

$$\left(A^{-1}\right)^T = \begin{pmatrix} D_t r & D_x r \\ D_t s & D_x s \end{pmatrix} = \begin{pmatrix} -2e^{-\kappa_1 s}\kappa_1 r^{3/2} & e^{-2\kappa_1 s}r \\ 0 & \frac{e^{-2\kappa_1 s}}{2\kappa_1} \end{pmatrix}.$$

Therefore, from

$$\begin{pmatrix} T_4^r \\ T_4^s \end{pmatrix} = J\left(A^{-1}\right)^T \begin{pmatrix} T_4^t \\ T_4^x \end{pmatrix}, \tag{28}$$

where $J = \det(A) = -\frac{e^{3\kappa_1 s}}{r^{3/2}}$, we obtain

$$T_4^r = \delta_1\kappa_1 + 2\delta_3\kappa_1 + \kappa_1 r^2 w_r^2 + \kappa_1 r w w_r + \frac{\kappa_1 w^2}{4} - \frac{1}{2}r^{3/2}ww_r^2 - \frac{w^3}{8\sqrt{r}} - \frac{1}{2}\sqrt{r}w^2 w_r,$$

$$T_4^s = -\frac{\delta_1 w_r}{w} - \frac{w^3}{16\kappa_1 r^{3/2}} - \frac{w^2 w_r}{4\kappa_1 \sqrt{r}} - \frac{\sqrt{r}ww_r^2}{4\kappa_1}. \tag{29}$$

From the reduced conservation law $D_r T_4^r = 0$, we obtain

$$\kappa_1 r^2 w_r^2 + \kappa_1 r w w_r + \frac{\kappa_1 w^2}{4} - \frac{1}{2}r^{3/2}ww_r^2 - \frac{w^3}{8\sqrt{r}} - \frac{1}{2}\sqrt{r}w^2 w_r = k,$$

where $k$ is an arbitrary constant.

*4.3. Double Reduction of ([1](1)) by $\left\langle \kappa_1\left(X_1 - \frac{X_3}{2}\right) + \kappa_2 X_2 \right\rangle$*

Canonical coordinates determined from $\left\langle \kappa_1\left(X_1 - \frac{X_3}{2}\right) + \kappa_2 X_2 \right\rangle$ are

$$r = x(2\kappa_2 - \kappa_1 t), \quad s = \frac{2\ln x}{\kappa_1}, \quad w = \frac{u}{x^2}, \quad \kappa_1 \neq 0, \tag{30}$$

where $w = w(r)$, and the inverse canonical coordinates are given by

$$t = \frac{2\kappa_2 - re^{-\frac{1}{2}\kappa_1 s}}{\kappa_1}, \quad x = e^{\frac{\kappa_1 s}{2}}, \quad u = we^{\kappa_1 s} \tag{31}$$

Therefore, the partial derivatives of $u$ from ([31](31)) are given by

$$\begin{aligned}
u_t &= -\kappa_1 w_r e^{\frac{3\kappa_1 s}{2}}, \quad u_x = e^{\frac{\kappa_1 s}{2}}(rw_r + 2w), \\
u_{tx} &= -\kappa_1 e^{\kappa_1 s}(rw_{rr} + 3w_r), \\
u_{xx} &= r(rw_{rr} + 4w_r) + 2w.
\end{aligned} \tag{32}$$

Therefore,

$$A = \begin{pmatrix} D_r t & D_r x \\ D_s t & D_s x \end{pmatrix} = \begin{pmatrix} -\frac{e^{-\frac{1}{2}\kappa_1 s}}{\kappa_1} & 0 \\ \frac{1}{2}e^{-\frac{1}{2}\kappa_1 s}r & \frac{1}{2}e^{\frac{\kappa_1 s}{2}}\kappa_1 \end{pmatrix}$$

and

$$\left(A^{-1}\right)^T = \begin{pmatrix} D_t r & D_x r \\ D_t s & D_x s \end{pmatrix} = \begin{pmatrix} -e^{\frac{\kappa_1 s}{2}}\kappa_1 & e^{-\frac{1}{2}\kappa_1 s}r \\ 0 & \frac{2e^{-\frac{1}{2}\kappa_1 s}}{\kappa_1} \end{pmatrix}.$$

Therefore, from

$$\begin{pmatrix} T_5^r \\ T_5^s \end{pmatrix} = J\left(A^{-1}\right)^T \begin{pmatrix} T_5^t \\ T_5^x \end{pmatrix}, \tag{33}$$

where $J = \det(A) = -\frac{1}{2}$, we obtain

$$T_5^r = \frac{2\kappa_1(2\delta_1 r w_r + 4\delta_1 w + \delta_3 r w_r + 2\delta_3 w - 1) - 2\delta_2(r w_r + 2w) - r(3r w_r + 4w)}{4r w_r + 8w},$$

$$T_5^s = -\frac{2\delta_1\kappa_1 r^2 w_r^2 + 4\delta_1\kappa_1 r w w_r + 2\delta_2 r w w_r + 4\delta_2 w^2 + 3r^2 w w_r + 4r w^2}{2\kappa_1 r^2 w w_r + 4\kappa_1 r w^2}. \tag{34}$$

From the reduced conservation law $D_r T_5^r = 0$, we obtain

$$\frac{2\kappa_1(2\delta_1 r w_r + 4\delta_1 w + \delta_3 r w_r + 2\delta_3 w - 1) - 2\delta_2(r w_r + 2w) - r(3r w_r + 4w)}{4r w_r + 8w} = k,$$

where $k$ is an arbitrary constant.

*4.4. Double Reduction of (1) by $\langle X_3 \rangle$*

Canonical coordinates determined from $X_3$ are

$$r = \frac{x}{t}, \quad s = \ln x \quad w = u, \tag{35}$$

where $w = w(r)$, and the inverse canonical coordinates are given by

$$t = \frac{e^s}{r}, \quad x = e^s \quad u = w. \tag{36}$$

Therefore, the partial derivatives of $u$ from (36) are given by

$$u_t = -r^2 e^{-s} w_r, \quad u_x = r e^{-s} w_r,$$

$$u_{tx} = -r^2 e^{-2s}(r w_{rr} + w_r), \tag{37}$$

$$u_{xx} = r^2 e^{-2s} w_{rr}.$$

As for $A$ and $\left(A^{-1}\right)^T$, we obtain

$$A = \begin{pmatrix} D_r t & D_r x \\ D_s t & D_s x \end{pmatrix} = \begin{pmatrix} -\frac{e^s}{r^2} & 0 \\ \frac{e^s}{r} & e^s \end{pmatrix},$$

and

$$\left(A^{-1}\right)^T = \begin{pmatrix} D_t r & D_x r \\ D_t s & D_x s \end{pmatrix} = \begin{pmatrix} -e^{-s} r^2 & e^{-s} r \\ 0 & e^{-s} \end{pmatrix}.$$

Therefore, from

$$\begin{pmatrix} T_3^r \\ T_3^s \end{pmatrix} = J\left(A^{-1}\right)^T \begin{pmatrix} T_3^t \\ T_3^x \end{pmatrix}, \tag{38}$$

where $J = \det(A) = -\frac{e^{2s}}{r^2}$, we obtain

$$T_3^r = \delta_2 - \delta_1,$$

$$T_3^s = \frac{1}{2} w_r^2(w - r) - \frac{\delta_1}{r} - w_r \phi_1(w). \tag{39}$$

It is remarkable that in this case, because $T_3^r$ in (39) is simply a constant, the reduced conservation law $D_r T_3^r = 0$ does not result in an ODE that can be solved for $w$. Therefore, no invariant solution arises via the double reduction method from the association of $X_3$ and the conservation law $T_3$.

## 5. Concluding Remarks

In this paper, a study of the Hunter–Saxton equation using Lie symmetry analysis was presented. Symmetry reductions of the equation were carried out by employing the generalized approach to double reduction theory proposed by Bokhari et al. [7]. By utilizing the multiplier method, nontrivial conservation laws for the Hunter–Saxton equation were derived. These conservation laws, along with the Lie point symmetries of the equation, were employed to perform symmetry reductions via the double reduction method.

Through the analysis, a set of first-order ODEs was obtained, whose solutions represent invariant solutions for the Hunter–Saxton equation. Out of the five nontrivial conservation laws constructed, it was observed that only four had associated Lie point symmetries according to the definition provided by Kara and Mahomed [16]. The conservation law $T_1$ did not have any linear combination of symmetries associated with it. Additionally, it is noteworthy that despite the conservation law $T_3$ having an associated Lie point symmetry, $X_3$, the application of the double reduction method in this case did not yield a symmetry reduction of the Hunter–Saxton equation. This outcome could be attributed to the "collapse" of the first integral, which was expected to represent a reduced ODE for the PDE but instead resulted in a constant value.

**Author Contributions:** Conceptualization, M.C.K. and W.S.; methodology, M.C.K., W.S. and P.M.; software, M.C.K. and W.S.; validation, W.S. and P.M.; formal analysis, M.C.K., W.S. and P.M.; writing—original draft preparation, M.C.K. and W.S.; writing—review and editing, M.C.K., W.S. and P.M. All authors have read and agreed to the published version of the manuscript.

**Funding:** This research received no external funding.

**Acknowledgments:** The authors would like to thank the Directorate of Research Development and Innovation of Walter Sisulu University for continued support.

**Conflicts of Interest:** The authors declare no conflict of interest.

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
