# Peer review of "Conservation Laws and Symmetry Reductions of the Hunter–Saxton Equation via the Double Reduction Method"

_mca, doi:10.3390/mca28050092_

Round 1

Reviewer 1 Report

please see the pdf file (commemt).

Please check for possible grammar and printing errors.

Author Response

We are grateful for the very useful comments from the reviewer.

Reviewer 2 Report

Comments on the manuscript mca-2525199-peer-review-v1 entitled

“Conservation Laws and Symmetry Reductions of the Hunter-Saxton Equation via the Double Reduction Method”

This manuscript uses the double reduction approach to carry out symmetry reductions of a nonlinear hyperbolic PDE in mathematical physics called the Hunter–Saxton equation.

By using the multiplier method, nontrivial conservation laws for the Hunter-Saxton equation have been derived. The derived conservation laws, along with the Lie point symmetries of the equation, have been employed to perform symmetry reductions via the double reduction method. Five nontrivial conservation laws of the Hunter-Saxton equation are derived, four of which are found to have associated Lie point symmetries.

Some suggestions and comments are given in below.

Minor comments

1.      The abstract should have modified by adding the advantages and dis-advantages of the proposed method to solve the Hunter-Saxton equation.

2.      The motivation of the manuscript is not clear enough. The authors failed to clearly explain why such a method is needed and why is it necessary to solve such an equation?

3.      Considering my second comment, please describe the real-world applications of nonlinear model (1) deeply and exactly.

4.      Page5, line 109, after \delta_i, please add the variations of i, i.e. \delta_i, i=1,2,..,5.

5.      Page5, line 113, after (T_i^1, T_i^2), please add the variations of i, i.e. i=1,2,..,5.

6.       In general, equations should be considered parts of a sentence and treated accordingly with the appropriate grammatical convention and punctuation. This point has been considered for most of equations, except equations 4, 5, the formulas below lines 84, 86, 88, and some others.

Author Response

(The authors gave the same response as above.)

Round 2

Reviewer 1 Report

This version is recommended for publication.